# Electroosmotic Flow of Viscoelastic Fluid through a Constriction Microchannel

**DOI:** 10.3390/mi12040417

**Published:** 2021-04-09

**Authors:** Jianyu Ji, Shizhi Qian, Zhaohui Liu

**Affiliations:** 1Department of Mechanical and Aerospace Engineering, Old Dominion University, Norfolk, VA 23529, USA; jji016@odu.edu; 2State Key Laboratory of Coal Combustion, School of Energy and Power Engineering, Huazhong University of Science and Technology, Wuhan 430074, China; zliu@mail.hust.edu.cn

**Keywords:** electroosmosis, microfluidics, elastic instability, non-Newtonian fluid, Oldroyd-B model

## Abstract

Electroosmotic flow (EOF) has been widely used in various biochemical microfluidic applications, many of which use viscoelastic non-Newtonian fluid. This study numerically investigates the EOF of viscoelastic fluid through a 10:1 constriction microfluidic channel connecting two reservoirs on either side. The flow is modelled by the Oldroyd-B (OB) model coupled with the Poisson–Boltzmann model. EOF of polyacrylamide (PAA) solution is studied as a function of the PAA concentration and the applied electric field. In contrast to steady EOF of Newtonian fluid, the EOF of PAA solution becomes unstable when the applied electric field (PAA concentration) exceeds a critical value for a fixed PAA concentration (electric field), and vortices form at the upstream of the constriction. EOF velocity of viscoelastic fluid becomes spatially and temporally dependent, and the velocity at the exit of the constriction microchannel is much higher than that at its entrance, which is in qualitative agreement with experimental observation from the literature. Under the same apparent viscosity, the time-averaged velocity of the viscoelastic fluid is lower than that of the Newtonian fluid.

## 1. Introduction

Electroosmotic flow (EOF) uses electric field to control fluid motion, and has been widely used in various microfluidic and nanofluidic applications such as fluid pump [1], mixing [2], and polymer translocation in biosensing [3]. The existing studies of EOF have been mainly focusing on Newtonian fluids [4,5]. However, in reality, EOF has been widely used to control and manipulate biological fluids (e.g., blood, saliva, lymph, protein, and DNA solutions) [6,7,8] and polymeric solutions [9], which exhibit non-Newtonian characteristics. Therefore, investigating EOF of viscoelastic fluids is of practical importance.

Bello et al. [10] conducted the pioneering study on EOF of non-Newtonian fluid, and measured EOF velocity of methyl cellulose solution in a capillary. Their results show that EOF velocity of such polymer solutions is much higher than that predicted with the classic Helmholtz-Smoluchowski velocity. Chang and Tsao [11] conducted similar experiments and found the effective viscosity decreased because of the sheared polymeric molecules inside the electrical double layer (EDL). Theoretically, non-Newtonian effects can be characterized by proper constitutive models relating the dynamic viscosity and the rate of shear. Such constitutive models include power-law model [12], Carreau model [13], WhiteMetzner model [14], Bingham model [15], Oldroyd-B (OB) model [16], PTT model [17], Moldflow second-order model [18], Giesekus model [19], etc. Das [20] developed an approximate solution for EOF velocity of power-law fluid between two parallel plates. Zhao et al. [21,22] derived a generalized Helmholtz–Smoluchowski velocity for EOF of power-law fluid in a slit microchannel. Later, Zhao and Yang [23,24] extended the study to a cylindrical microcapillary. Olivares et al. [25] experimentally investigated EOF of a non-Newtonian polymeric solution and verified the generalized Helmholtz–Smoluchowski velocity. Tang et al. [26] numerically investigated EOF of power-law fluid using Lattice–Boltzmann method. Zimmerman et al. [13] carried out numerical simulation of EOF of Carreau fluid in a T-junction microchannel, and found that the flow field significantly depended on the non-Newtonian characteristics of the fluid. The aforementioned studies on EOF of non-Newtonian fluid are limited to inelastic constitutive models (i.e., power-law and Carreau models). However, some fluids show both viscous and elastic behaviors, which can be presented by viscoelastic constitutive models. There are existing literatures investigating the characteristics of EOF of viscoelastic fluids [27,28,29,30,31], showing that the viscoelasticity of the fluid affects the flow pattern and flow rate. Note that in the aforementioned studies, the EOF of non-Newtonian fluid was assumed in a steady state.

Recently, EOFs of non-Newtonian fluids have been reported to be time-dependent and show instabilities even at low Reynolds number (*Re*). Such EOFs are time-dependent because of the nonlinear viscosity and elasticity of non-Newtonian fluids. Bryce and Freeman [32] first reported the electro-elastic instability in EOF of PAA solutions through a 2:1:2 micro-scale contraction/expansion when the applied electric field exceeded a threshold value. Later, Bryce and Freeman [33] reported that such instabilities insignificantly enhanced the mixing in micro flows. Pimenta and Alves [34,35] later experimentally and numerically studied the electro-elastic instabilities of PAA solutions in both cross-slot and flow-focusing micro devices, and found that mixing efficiency was not improved significantly. Song et al. [36] experimentally and numerically studied the elastic instability in EOF of viscoelastic polyethylene oxide (PEO) solutions through T-shaped microchannels, and results demonstrated that the threshold electric field for onset of instability highly depended on the PEO concentration. Song et al. [37] later extended the work by experimentally investigating the fluid rheological effects on the elastic instability in EOF of six types of phosphate buffer-based aqueous solutions through T-shaped microchannels. They found that shear thinning effect of the fluid might account for the electro-elastic instabilities, however, the fluid with high elasticity alone did not have instability, which is inconsistent with the results of Pimenta [35]. The authors attribute the inconsistency to the neglect of microstructural effects (e.g., polymer-wall interaction and electric effect on molecular structure of polymer, etc.) of shear-thinning polymer solutions. However, this experimental result shows similarity to the work of Ko et al. [38], in which weakly shear-thinning, viscoelastic polyvinylpyrrolidone (PVP), and PEO solutions exhibited Newtonian-like EOF patterns, while shear-thinning and weakly elastic xanthan gum (XG) solution exhibited disturbance and vortices, suggesting that fluid elasticity alone has an insignificant impact on the steady-state EOF pattern. More recently, Sadek [39] experimentally investigated EOF of viscoelastic fluids through different microchannel configurations, including hyperbolic-shaped contractions followed by an abrupt expansion, and abrupt contractions followed by a hyperbolic-shaped expansion, and EOF showed instabilities of elastic origin at very low Weissenberg numbers (*Wi*) (i.e., *Wi* < 0.01).

There is only limited literature on numerical studies of electro-elastic instabilities. Afonso et al. [40] numerically investigated the elastic instability of EOF through a cross-slot geometry using the upper-converted Maxwell and the simplified Phan-Thien-Tanner models, and a direct flow transition from steady symmetric state to unsteady flow without crossing the steady asymmetric state at a critical *Wi* was observed. Pimenta and Alves [35] numerically investigated the electro-elastic instabilities in cross-slot and flow-focusing micro devices using OB model and Poisson-Boltzmann (PB) model. They found that strong shear-dominated flow within the EDL at the corners had a more significant contribution to the elastic instabilities than the extensionally dominated bulk flow. Song et al. [36] numerically investigated EOF of PEO solution through a T-shaped microchannel. Their model considered only the influence of PEO solution on the fluid viscosity, conductivity, and zeta potential. Due to the neglect of fluid elasticity effect in the mathematical model, only the electrokinetic flow phenomena of dilute PEO solution (i.e., ≤750 ppm) were captured.

Both experimental and numerical investigations in the EOF instabilities of viscoelastic fluid are limited, and the conditions proposed by various researchers for triggering the instabilities in the EOF of viscoelastic fluids show inconsistency and remain unclear. Inspired by the existing literature, in this work we numerically study EOF of viscoelastic fluids through a 10:1:10 contraction microchannel. The geometry, which consists of a constriction microchannel connecting two relatively big reservoirs on either end, is close to actual microfluidic device. The time-dependent OB model and PB model are adopted to describe the constitutive characteristics and the electrokinetic phenomenon, respectively. EOFs of PAA solutions with various weight concentrations under different applied electric fields are investigated. The effects of polymer concentration and applied electric field on the elastic instability are studied, and a map in polymer concentration-electric field space for predicting the onset of upstream vortices is formed.

## 2. Mathematical Model

We consider incompressible monovalent binary electrolyte solution such as KCl with bulk concentration $c_{0}$ mixed with PAA polymer solution of concentration $c_{p}$, which fills a microchannel of height $H_{c}$, length $L_{c}$, and width *W* connecting two identical reservoirs of height $H_{r}$ and length $L_{r}$ on either side. The solid walls of the constriction microchannel and the reservoirs are assumed to carry a constant negative zeta potential, $\xi_{0}$. When dealing with non-Newtonian fluids, a constant zeta potential has been widely accepted [41]. Huang et al. [41] compared theoretical and experimental results of PEO solutions, and a constant zeta potential was proven for various PEO concentrations. Therefore, in the current study, we neglect the effect of the polymer concentration on the wall zeta potential. Two electrodes are placed at both ends of the reservoirs, and an external potential bias $U_{0}$ is applied between the inlet (Anode) and outlet (Cathode). Through the interaction between the externally applied electric field and net charges accumulated within the EDL in the vicinity of the charged walls, EOF flowing from the anode reservoir through the constriction microchannel towards the cathode reservoir is generated. The apparent electric field between the inlet and outlet is defined as $E_{\mathrm{app}}=U_{0}/\left(2L_{r}+L_{c}\right)$. In some applications, there are slit microchannels with width much larger than height [42,43]. For example, two-phase flow patterns were studied in a microchannel with 10-mm width and 50-μm height [42]. For microchannels with such geometries, the flow can be simplified to a 2D problem [44]. Therefore, in the current study we assume that the channel width is much larger than the channel height, and the flow can be simplified to a 2D problem as schematically shown in Figure 1. A Cartesian coordinate system with origin fixed at the center of the microchannel is adopted with the *x*-axis along the length direction and the *y*-axis in the height direction.

The induced EOF of viscoelastic fluid is governed by the continuity and Navier–Stokes equations:(1)∇·u=0,
(2)$$\rho \left(\frac{\partial u}{\partial t}+u\cdot \nabla u\right)=-\nabla p+\eta_{s}\nabla^{2}u+\nabla \cdot \tau -\rho_{E}\nabla \phi_{\mathrm{Ext}},$$
where ***u*** is the velocity; *p* is the pressure; *ρ* is the fluid density; *η_s_* denotes the solvent dynamic viscosity; τ is the polymeric stress tensor accounting for the memory of the viscoelastic fluid; *t* represents time; $\rho_{E}$ and $\phi_{\mathrm{Ext}}$ represent, respectively, the volume charge density within the electrolyte solution and the externally applied electric potential. For different types of viscoelastic fluids, various constitutive models have been developed to relate the polymeric stress tensor ***τ*** and the deformation rate of the fluid, including WhiteMetzner model [14], which is commonly used for shear-thinning fluid; PTT model [17], which has good performance for prediction of viscosity at low shear rates; Giesekus model [19], which is suitable for concentrated polymer solutions; and OB model [16], which is suitable for dilute polymer solutions. Since OB model can properly fit the rheological behavior of aqueous PAA solutions [45], OB model is adopted in this study. In the OB model, the polymeric stress tensor, ***τ***, is described as [16],
(3)$$\tau =\frac{\eta_{p}}{\lambda }\left(c-I\right),$$
where *η*_p_ is the polymer dynamic viscosity; *λ* is the relaxation time of the polymer, which refers to the time it takes for polymer chains to return to equilibrium after being disturbed; ***c*** is the symmetric conformation tensor of the polymer molecules; and ***I*** is the identity matrix.

For the OB model, the conformation tensor ***c*** is governed by [16],
(4)$$\frac{\partial c}{\partial t}+u\cdot \nabla c=c\cdot \nabla u+{\left(\nabla u\right)}^{T}\cdot c-\frac{1}{\lambda }\left(c-I\right).$$

Typically, numerical simulation of viscoelastic flow is difficult to converge for high Weissenberg number problem [46,47]. Computations were found to break down at frustratingly low values of Weissenberg number (usually around *Wi* = 1; precise critical value also depends on the flow geometry) [48]. Therefore, the log-conformation tensor approach [47] is adopted. In the log-conformation tensor method, a new tensor (Θ) is defined as the natural logarithm of the conformation tensor,
(5)Θ=ln(c)=Rln(Λ)R,
where Λ is a diagonal matrix whose diagonal elements are the eigenvalues of ***c***; and **R** is an orthogonal matrix with its columns being the eigenvalues of ***c***. Equation (4) for the conformation tensor written in terms of Θ then becomes [46],
(6)$$\frac{\partial \Theta }{\partial t}+u\cdot \nabla \Theta =\Omega \Theta -\Theta \Omega +2B+\frac{1}{\lambda }\left(e^{\Theta }-I\right).$$
In the above, Ω and B are, respectively, the anti-symmetric matrix and the symmetric traceless matrix of the decomposition of the velocity gradient tensor ∇u [46].

Then, the conformation tensor ***c*** is recovered from Θ,
(7)c=exp(Θ).

The total electric potential, Ψ, is decomposed in two variables, $\Psi =\phi_{\mathrm{Ext}}+\psi$ [35], with $\phi_{\mathrm{Ext}}$ representing the potential originated from the externally applied electric potential while ψ being the potential arising from the charge of channel walls. In this study, the EDL thickness is on the order of nanometers (the calculation of EDL thickness will be shown in the next paragraph), while the microchannel height is on the order of micrometers. Therefore, the Poisson–Boltzmann equation [49] is used to describe the potential, ψ:(8)$$\nabla \cdot \left(\varepsilon \nabla \psi \right)=\rho_{E}=Fc_{0}\left(\exp\left(\frac{e\psi }{kT}\right)-\exp\left(-\frac{e\psi }{kT}\right)\right).$$

In the above, *F* is the Faraday’s constant (i.e., 96,485.33289 $C\cdot {\mathrm{mol}}^{-1}$); *e* is the elementary charge (i.e., $1.6021766341\times 10^{-19}\;C$); *k* is Boltzmann’s constant (i.e., $1.380649\times 10^{-23}\;J\cdot K^{-1}$); *T* is the absolute temperature of the fluid (i.e., 295 K); and ε represents the permittivity of the solution (i.e., $6.906266\times 10^{-10}\;F\cdot m^{-1}$). In this study, the bulk concentration $c_{0}$ is 0.01 mM; $z_{1}=1$ and $z_{2}=-1$. In a biocompatible solution with pH of 7.4, the concentration of $H^{+}$ is $10^{-7.4}\;\mathrm{mol}/L$, and the concentration of ${\mathrm{OH}}^{-}$ is $10^{-6.6}\;\mathrm{mol}/L$. The concentration of weak electrolyte is relatively low comparing with the background salt. Therefore, the weak electrolyte is not considered in the current study. The EDL thickness can be calculated by $\lambda_{D}=\sqrt{\frac{\varepsilon kT}{eF\left(z_{1}^{2}c_{0}+z_{2}^{2}c_{0}\right)}}$, which is 95 nm. The potential $\phi_{\mathrm{Ext}}$ is governed by the following Laplace equation [50],
(9)$$\nabla^{2}\phi_{Ext}=0.$$

The boundary conditions are given as follows (Figure A4):

(1) At the Anode (edge AG in Figure A4): n·∇u=0; p=0; τ=0; $\phi_{\mathrm{Ext}}=U_{0}$; n·∇ψ=0; Θ=0; where **n** denotes the normal unit vector on the surface.

(2) At the Cathode (edge FL in Figure A4): n·∇u=0; p=0; n·∇τ=0; $\phi_{\mathrm{Ext}}=0$; n·∇ψ=0; n·∇Θ=0.

(3) On the reservoir walls (edges ABC, DEF, GHI, and JKL in Figure A4) and the microchannel walls (edges CD and IJ in Figure A4): u=0; $n\cdot \nabla \phi_{\mathrm{Ext}}=0$; $\psi =\xi_{0}$; n·∇Θ=0; n·∇p is obtained from the momentum equation; the components of τ are linearly extrapolated.

The following initial conditions are specified within the domain: u=0; p=0; τ=0; $\phi_{\mathrm{Ext}}=0$; ψ=0; Θ=0.

Note that the electric potentials and the flow are only one-way coupling. The electric potentials $\phi_{\mathrm{Ext}}$ and ψ are in a steady state, and they are independent on the flow. However, the electric potentials affect the flow through the electrostatic force, which is the last term in Equation (2). For Newtonian fluid, the third term, ∇·τ, in the right-hand-side of Equation (2) is dropped, and the model includes Equations (1), (2), (8), and (9).

## 3. Numerical Method and Code Validation

The governing equations are numerically solved using the finite volume method by RheoTool (version 4.1, https://github.com/fppimenta/rheoTool, accessed on 1 June 2020), an open-source viscoelastic EOF solver [35] implemented in the open-source OpenFOAM platform. The details of the solver can be found from the work of Pimenta and Alves [34,50]. To numerically solve the coupled Equations (1), (2), (6), (8), and (9) along with the boundary and initial conditions, CUBISTA scheme [51] is used to discretize the convective terms in Equations (2) and (6). Central differences are used for the discretization of Laplacian and gradient terms. The time derivatives are discretized with three-time level explicit difference scheme [52], which is of the second order of accuracy. The exponential source term in Equation (8) is linearized using Taylor expansion up to the second term [53]. All of the terms in the momentum equation (i.e., Equation (2)), except the pressure gradient and the electric contribution, are discretized implicitly. A small time-step, $\Delta t=\lambda /10^{5}$, is used to ensure the accuracy. The well-known SIMPLEC (Semi-Implicit Method for Pressure-Linked Equations-Consistent) algorithm [54] is used to resolve the velocity-pressure coupling. An inner-iteration loop is used to reduce the explicitness of the method and increase its accuracy and stability. The pressure field is computed by PCG (Preconditioned Conjugate Gradient) solver, of which the tolerance and maximum iteration are set to be $1\times 10^{-8}$ and 800, respectively. The velocity field is computed by PBiCG (Preconditioned Biconjugate Gradient) solver, of which the tolerance and the maximum iteration are set to be $1\times 10^{-10}$ and 1000, respectively. The computational steps of the solver are as follows [34]:
Step 1. Initialize the fields ${\left\{u,p,\tau ,\phi_{\mathrm{Ext}},\psi ,\Theta \right\}}_{0}$ and time (t=0).
Step 1.1. Compute steady state $\phi_{\mathrm{Ext}}$ from Equation (9) and ψ from Equation (8).Step 2. Enter the time loop (t=Δt).
Step 2.1. Enter the inner iteration loop (i=0).
Step 2.1.1. Compute $\Theta_{i}$ and $\tau_{i}$ by log-conformation method.Step 2.1.2. Compute estimated velocity field $u_{i}^{*}$ by solving the momentum equation.Step 2.1.3. ompute pressure field $p_{i}$ by enforcing the continuity equation.Step 2.1.4. Correct the previously estimated velocity field using the correct pressure field.Step 2.1.5. Increase the inner iteration index (i=i+1) and repeat the computation from Step 2.1.1, until the inner iteration criteria (i.e., maximum tolerance) is satisfied.Step 2.1.6. Set ${\left\{u,p,\tau ,\phi_{\mathrm{Ext}},\psi ,\Theta \right\}}_{t}=\left\{u_{i},p_{i},\tau_{i},\phi_{\mathrm{Ext}}{}_{i},\psi_{i},\Theta_{i}\right\}$.Step 2.2. Increase time, t=t+Δt, and return to Step 2.1 until the simulation time is reached.Step 3. Stop the simulation and exit.

Structural mesh is adopted to discretize the computational domain. 90° corners of the contraction channel (points I, J, C, and D in Figure 1) are smoothed by a fillet of 1 μm in radius to avoid sharp turns. The 90° corners of the reservoirs (points H, K, B, and E in Figure 1) are smoothed by a fillet of 2 μm in radius. To capture the EDL in the vicinity of the charged walls, a finer mesh is distributed near the charged reservoir and channel walls as shown in Figure 2. To reduce the number of mesh, we use a relatively low bulk concentration $c_{0}=0.01\;\mathrm{mM}$, and the EDL thickness is 95 nm in this study. In order to capture the details in the EDL and to guarantee the accuracy, the mesh size near the charged wall is set to be 10 nm so that there are 10 meshes within the EDL. There are 77,192 meshes in the whole geometry. A mesh independence study, described in the Appendix B, is performed to ensure the accuracy of the simulation.

In this work, $\eta_{p}$ and λ for 100 ppm, 250 ppm, and 1000 ppm PAA-water solutions [55] are adopted to accomplish curve fitting as shown in Figure 3. The values of $\eta_{p}$ and λ were experimentally measured [56], and the slow retraction method was used to measure the relaxation time. The polymer dynamic viscosity can be expressed as $\eta_{p}=2.22\times 10^{-5}\cdot c_{p}$*,* and the relaxation time can be expressed as $\lambda =3.69\times 10^{-3}+3.94222\times 10^{-5}\cdot c_{p}+9.68889\times 10^{-5}\cdot c_{p}{}^{2}$, where $c_{p}$ represents the weight concentration of PAA solution with the unit of ppm. The $\eta_{p}$ and λ for other $c_{p}$ studied in this work are estimated by the curve-fitting expressions.

In a microfluidic channel with EDL thickness much smaller than the channel height, the EOF velocity of a Newtonian fluid can be approximated by the Helmholtz–Smoluchowski velocity formula [57],
(10)$$u_{0}=-\frac{\varepsilon \xi_{0}E_{x}}{\eta_{0}},$$
where $E_{x}$ is the actual local electric field in the main stream direction, and $\eta_{0}$ is the total viscosity of the fluid. To check the accuracy of our code, we simulate EOFs of both Newtonian and viscoelastic fluids in the same geometry with $H_{c}=40\;\mu m$, $L_{c}=200\;\mu m$, $H_{r}=400\;\mu m$, and $L_{r}=400\;\mu m$. Other parameters are set as $U_{0}=60\;V$,$\;\xi_{0}=-0.11\;V$ [58], and $\varepsilon =6.906266\times 10^{-10}\;F\cdot m^{-1}$. For Newtonian fluid, the total viscosity is set as $\eta_{0}=\eta_{s}=0.00322\;\mathrm{kg}/\left(m\cdot s\right)$. When the concentration of PAA solution is less than 2 ppm, the relaxation time is less than 0.1 ms [56], and the fluid can be approximately treated as Newtonian fluid. Therefore, for the OB model, parameters are set as $\eta_{s}=0.00317\;\mathrm{kg}/\left(m\cdot s\right)$, $\eta_{p}=0.00005\;\mathrm{kg}/\left(m\cdot s\right)$, $\eta_{0}=\eta_{s}+\eta_{p}=0.00322\;\mathrm{kg}/\left(m\cdot s\right)$, and λ=0.1 ms. Figure 4a depicts electric potential $\phi_{\mathrm{Ext}}\left(x,0\right)$ along the *x*-axis when $E_{\mathrm{app}}=600\;V/\mathrm{cm}$. The electric field in the *x*-direction, $-\frac{\partial \phi_{\mathrm{Ext}}}{\partial x},$ in the constriction microchannel is 1820 V/cm, which is about 10 times of the electric field in the reservoirs. This is because of the 10:1:10 contraction geometry and current conservation. With the same electric conductivity, the electric field is inversely proportional to the cross-sectional area of the geometry. Note that the actual electric field within the constriction microchannel is about three times of the apparent electric field, $E_{\mathrm{app}}$, which does not consider the cross-sectional variation of the geometry. EOFs of both Newtonian fluid and viscoelastic fluid reach a steady state. Figure 4b shows the *x*-component velocity profiles, u(0,y), of the Newtonian fluid (solid line) and the OB model (circles). The velocity first rises rapidly within the thickness of EDL, then reaches a plateau in the cross section of the channel. When $E_{x}=1820\;V/\mathrm{cm}$, the calculated Helmholtz–Smoluchowski velocity is 4.29 mm/s, and the velocity at the center of the channel is 4.27 mm/s for both Newtonian and OB models. The relative difference between the approximated velocity and the simulated velocity is less than 0.5%. In addition, the result for OB model matches that of Newtonian fluid. Such consistency between Newtonian model and OB model is because the polymer dynamic viscosity $\eta_{p}$ is much smaller than the solvent dynamic viscosity $\eta_{s}$, and the relaxation time of the polymer λ is also tiny. Under the considered condition, the elastic effect of the fluid is negligible and the OB fluid is almost the same as Newtonian fluid with the same total viscosity.

Afonso et al. [28] derived an analytical solution of viscoelastic EOF between two parallel plates based on the Debey–Hückel approximation, which is valid under the condition of low zeta potential (i.e., $\xi_{0}<25\;\mathrm{mV}$). To further validate our code for OB model, EOF of viscoelastic fluid with $\eta_{s}=0.001\;\mathrm{kg}/\left(m\cdot s\right)$, $\eta_{p}=0.00222\;\mathrm{kg}/\left(m\cdot s\right)$, $\eta_{0}=\eta_{s}+\eta_{p}=0.00322\;\mathrm{kg}/\left(m\cdot s\right)$, and λ=8.6 ms in a straight 2D channel (with height of 40 μm) is studied. These rheology parameters are corresponding to those of 100 ppm PAA solution. $U_{0}$ is set as 10 V, while $\xi_{0}$ is chosen as −10 mV and −110 mV, respectively. Under the considered conditions, the flows are steady state due to relatively low electric field strength. Figure 5 depicts the *x*-component velocity profile at the center of the channel, and our numerical results (triangles) are in excellent agreement with the analytical result (line). Although the analytical solution is based on the Debey–Hückel approximation, we find that the numerical result also agrees well with the analytical solution when the zeta potential $\xi_{0}$ is −110 mV. Therefore, the agreement of results attained from the OB model and Newtonian model, which are also validated by the Helmholtz–Smoluchowski approximation, as well as the agreement of analytical solution of OB model and numerical results for EOF of viscoelastic fluid in a straight channel, validate our code.

## 4. Results and Discussion

Newtonian fluid is investigated to provide the reference flow characteristic for the contraction geometry. For the PAA solution with different concentration $c_{p}$, the applied apparent electric field $E_{\mathrm{app}}$ is varied from low values to high (i.e., 100–600 V/cm). In this section, first, we describe the flow pattern of Newtonian fluid and the time-dependent flow patterns of PAA solutions. Then, the instabilities of PAA solutions with various $E_{\mathrm{app}}$ and $c_{p}$ are discussed and a flow map is formed based on the investigated values of $E_{\mathrm{app}}$ and $c_{p}$. Finally, statistical results of cross-sectional average velocity are presented.

### 4.1. Instability of PAA Solutions

For Newtonian fluids with various total viscosities, the EOF reaches a steady state under all conditions of the applied electric field strengths. There is no vortex occurring in the reservoirs and the constriction microchannel. The streamlines of Newtonian fluid show excellent symmetry about the *x*-axis. Additionally, the magnitude of the velocity, U(x,y), is symmetric about the *y*-axis, U(x,y)=U(−x,y). For EOF of PAA solutions, when $E_{\mathrm{app}}$ and $c_{p}$ are relatively low, the flow pattern is similar to that of Newtonian fluid, and the flow reaches a steady state without vortex. With increasing $E_{\mathrm{app}}$ and $c_{p}$, however, the viscoelastic flow becomes time dependent and significant instabilities are observed. Figure 6 depicts the streamlines at different times when $E_{\mathrm{app}}=100\;V/\mathrm{cm}$ and $c_{p}=500\;\mathrm{ppm}$. Figure 7 depicts the streamlines at different times when $E_{\mathrm{app}}=600\;V/\mathrm{cm}$ and $c_{p}=150\;\mathrm{ppm}$. Figure 8 shows the velocity magnitudes as a function of time at three different locations, namely, upstream of the constriction microchannel $\left(-3H_{c},0\right)$, center of the constriction microchannel (0,0), and downstream of the constriction microchannel $\left(3H_{c},0\right)$. For the EOF of both $c_{p}=150\;\mathrm{ppm}$ and $c_{p}=500\;\mathrm{ppm}$, we observe strong instabilities and upstream vortices.

Figure 6 and Figure 7 show that the viscoelastic EOF is time-dependent. The streamlines in the left inlet reservoir far away from the solid walls (AB and GH in Figure 1) and near the entrance of the constriction microchannel show significant fluctuation and also become asymmetric about the *x*-axis. However, the streamlines near the solid walls of both reservoirs and in the outlet reservoir show insignificant change with time. Within 0.1 s, vortices continuously form and disappear within the inlet reservoir right before the entrance of the constriction microchannel. In Figure 6a, we observe significant curvature at the streamlines of the EOF upstream of the constriction microchannel. Then, the curvature of the streamlines further develops into a pair of vortices as shown in Figure 6b. Such vortices are also time-dependent. The size and shape of the vortices show notable differences at different times. After growing to the maximum size, the vortices start to shrink until the vortices break and disappear, as shown in Figure 6,d. Next, the vortices in the EOF keep forming and breaking repeatedly as shown in Figure 6e,f. Comparing Figure 6b,c, the central locations of the vortices are both spatially and temporally dependent. In Figure 6b, the direction of the circulation is marked by red curved lines. The pair of vortices are in opposite directions and form a stagnant region right before the entrance of the constriction microchannel. Therefore, we call the induced vortices as entrance-centerline vortices.

For $c_{p}=150\;\mathrm{ppm}$ and $E_{\mathrm{app}}=600\;V/\mathrm{cm}$, the width and length of the vortices are nearly the same as the height of the constriction microchannel ($H_{c}$), while for $c_{p}=500\;\mathrm{ppm}$ and $E_{\mathrm{app}}=100\;V/\mathrm{cm}$, the width and the length of the vortices are about $2H_{c}$. In the similar geometry, however, Ko [38] did not observe vortices in their experiments with 200 ppm PAA solution under $E_{\mathrm{app}}$ ranging from 75 V/cm to 200 V/cm. The elastic instability increases with increasing polymer concentration and the applied electric field. Our later results discussed in the following section show that for $c_{p}=200\;\mathrm{ppm}$, the vortices occur when the applied electric field exceeds the threshold value of 300 V/cm. Therefore, our numerical results qualitatively agree with the experimental observation of Ko [38] under their experimental condition. Table A1 in the Appendix B summarizes the EOF instabilities from the literature. In Sadek’s [39] experimental study, small vortices at the entrance and large upstream circulation flows were observed. For highly concentrated polymer solution, downstream circulation flows were observed at a critical voltage. The upstream vortices found in this study are distinct from the small vortices and large circulation flows found in Sadek’s study [39] in terms of location. Note that the geometry in our study differs significantly from the experimental study of Sadek [39], and we do not observe large circulating flows near the reservoir corners and channel lips.

For Newtonian fluid, the EOF reaches a steady state, and the velocity magnitudes are symmetric about the *y*-axis. Therefore, for Newtonian fluid, we have $U\left(-3H_{c},0\right)=U\left(3H_{c},0\right)$. However, as shown in Figure 8, the velocity magnitudes at three points $\left(-3H_{c},0\right)$, (0,0), and $\left(3H_{c},0\right)$ fluctuate around certain values and velocity magnitudes do not show symmetry about the *y*-axis. For $c_{p}=500\;\mathrm{ppm}$ and $E_{\mathrm{app}}=100\;V/\mathrm{cm}$, as shown in Figure 8b, the time-averaged velocity at the channel center is 0.201 mm/s with a standard deviation of 0.013 mm/s. The time-averaged velocity at downstream of the constriction microchannel is 0.108 mm/s, which is about 2 times of that at upstream of the constriction microchannel (i.e., 0.047 mm/s). For $c_{p}=150\;\mathrm{ppm}$ and $E_{\mathrm{app}}=600\;V/\mathrm{cm}$, the time-averaged velocities at the upstream, center, and downstream of the constriction microchannel are, respectively, 0.19 mm/s, 2.82 mm/s, and 2.05 mm/s. The ratio of the downstream velocity to upstream velocity is about 10 times. In contrast to Newtonian EOF, the flow velocity of viscoelastic fluid at the downstream is significantly higher than that at the upstream, which has also been experimentally observed in Ko’s [38] experiments, where a fluid jet after the constriction microchannel was observed and the ratio of the velocity at the downstream centerline to that at upstream of the constriction microchannel varies between 1 and 2 under $E_{\mathrm{app}}$ ranging from 75 V/cm to 200 V/cm and $c_{p}=200\;\mathrm{ppm}$.

Figure 8 also shows that EOF of $c_{p}=500\;\mathrm{ppm}$ and $E_{\mathrm{app}}=100\;V/\mathrm{cm}$ presents stronger instabilities than that of $c_{p}=150\;\mathrm{ppm}$ and $E_{\mathrm{app}}=600\;V/\mathrm{cm}$. Comparing Figure 6 and Figure 7, the streamlines show stronger fluctuation and larger upstream vortices for solution with relatively high polymer concentration. Such trend suggests that although the increase of both $E_{\mathrm{app}}$ and $c_{p}$ can enhance the instabilities of the viscoelastic EOF, the polymer concentration, $c_{p}$, affects the instabilities of the EOF more significantly, which will be further discussed in next section.

Figure 6 and Figure 7 also show the spatial distribution of elastic normal stress $\tau_{xx}$ with the color bar representing its magnitude. To clearly reveal it, Figure 9 depicts the spatial distribution of $\tau_{xx}$ in the whole geometry for $c_{p}=150\;\mathrm{ppm}$ and $E_{\mathrm{app}}=600\;V/\mathrm{cm}$ at t=1.78 s. Within the two reservoirs, the elastic normal stress is nearly zero at location far away from the constriction microchannel. However, significant elastic normal stress is induced near the entrance of the constriction microchannel and near the downstream lips. Due to the contraction geometry, the electric field within the constriction microchannel is about 10 times of that within the inlet reservoir as shown in Figure 4a, and the flow velocity in the microchannel is significantly higher than that in the reservoir. For example, Figure 8 shows that the ratio of the time-averaged velocity within the microchannel to that in the inlet reservoir, $U\left(0,0\right)/U\left(-3H_{c},0\right)$, is 4.28 for $c_{p}=500\;\mathrm{ppm}$ and $E_{\mathrm{app}}=100\;V/\mathrm{cm}$ and 10.79 for $c_{p}=150\;\mathrm{ppm}$ and $E_{\mathrm{app}}=600\;V/\mathrm{cm}$. Near the entrance of the microchannel, the high velocity gradient results in a strong extension of polymer molecules, and consequently induces significant elastic normal stress. Therefore, $\tau_{xx}$ experiences a rapid increase near the entrance of the constriction microchannel. At the exit of the constriction microchannel, similarly, a significant increase of $\tau_{xx}$ is induced at the exit lips. Figure 10a,b depict the streamlines in the constriction microchannel and the color bar represents the velocity magnitude, U, for $c_{p}=150\;\mathrm{ppm}$ and $E_{\mathrm{app}}=600\;V/\mathrm{cm}$ and Newtonian fluid with the same total viscosity and $E_{\mathrm{app}}$ at t=1.78 s, respectively. For the viscoelastic fluid, at both the entrance and exit of the constriction microchannel, as shown by the dashed circles in Figure 10a, velocity becomes spatially dependent along the *y*-axis. Velocity near the walls of the constriction microchannel is significantly higher than that at the centerline of the microchannel, and a local maximum occurs near the inlet/outlet corners of the constriction microchannel. However, in the EOF of Newtonian fluid, as shown in Figure 10b, at both the entrance and exit of the constriction microchannel, the velocity magnitude is more evenly distributed in the cross section of the constriction microchannel. Figure 10c depicts the velocity magnitude profile at the entrance ($2x/H_{c}=-5$) and exit ($2x/H_{c}=5$) of the constriction microchannel. For Newtonian fluid, the velocity magnitude profile is identical at $2x/H_{c}=\pm 5$ and is symmetric about the *x*-axis. The ratio of the maximum velocity magnitude near the channel walls to that at the centerline is 1.6. However, for PAA solution, due to the elastic instability, the velocity magnitude profile is asymmetric about the *x*-axis at $2x/H_{c}=\pm 5$. In addition, the ratios of the maximum velocity magnitude near the channel walls to that at the centerline are 9.7 and 4 at $2x/H_{c}=-5$ and $2x/H_{c}=5$, respectively, which are much higher than that of the Newtonian fluid. For Newtonian fluid, the velocity profile is symmetric about the centerline of the microchannel (i.e., *y* = 0). However, Figure 10c shows that the local maximum velocity near the top channel wall differs from that near the bottom channel wall, and the velocity profile is asymmetric about y=0.

As PAA solution flows from the microchannel into the outlet reservoir, fluid velocity first decreases when fluid exits the microchannel and then increases in the outlet reservoir, as shown by the region marked with a circle in Figure 10a and by the velocity magnitude as a function of *x* at y=0 in Figure 10d. The two dashed lines in Figure 10d represent the entrance and exit of the constriction microchannel. EOF of Newtonian fluid within the constriction is a plateau, and its velocity magnitude within the constriction is much higher than those at both reservoirs, and this is because the electric field within the constriction microchannel is significantly higher than that in the reservoirs. However, the velocity of PAA solution becomes spatially dependent within the constriction, and a local maximum occurs before the exit and a local minimum occurs at the exit of the constriction microchannel. In addition, a local maximum occurs at the downstream outlet reservoir. Figure 10d also clearly shows that the velocity in the downstream outlet reservoir is significantly higher than that at the upstream inlet reservoir. For example, $U\left(2x/H_{c}=10.0\right)/U\left(2x/H_{c}=-10.0\right)=3.37$. The unexpected velocity decrease at the microchannel exit and velocity increase at the downstream outlet reservoir do not occur in Newtonian fluid as shown in Figure 10b,d. Such a phenomenon is probably because of the extrudate swell effect of polymers [59]. At the exit of the constriction microchannel, curved streamlines tilting toward the walls of the constriction microchannel are observed in viscoelastic fluid, suggesting that fluid tends to flow toward the charged walls of the microchannel. Such lateral velocity component results in the velocity’s increase near the microchannel walls and velocity’s decrease near the centerline at the exit of the constriction microchannel. In addition, the significant increase of $\tau_{xx}$ near the downstream lips observed in Figure 9 can also be attributed to the extrudate swell effect of polymers when polymer exits from the constriction microchannel to larger outlet reservoir.

The generated elastic normal stress $\tau_{xx}$ was typically used to explain the formation of vortices in pressure-driven viscoelastic flows within curved geometries [60,61,62]. It has been reported that the development of the polymeric elastic stresses is caused by the flow-induced changes of the polymer conformation in the solution. Such changes of the polymer conformation are strain-dependent, anisotropic, and dependent on the flow. The extra elastic stresses are nonlinear under shear and can alter the flow behavior. At low Reynolds numbers where inertia is negligible, when the elastic normal stress exceeds by a certain amount the local shear stress, the flow transits from stable to unstable, and the vortices form at upstream of the constriction microchannel. Such elastic instabilities are often observed in flows with sufficient curvature [63,64,65], and some argue that curvature is necessary for infinitesimal perturbations to be amplified by the normal stress imbalances in the viscoelastic flows [66]. However, other theoretical studies reported that viscoelastic flows also showed a nonlinear instability in parallel shear flows, such as in viscoelastic flows within straight pipes at low Reynolds numbers [67]. Although the formation of the upstream vortices in the viscoelastic EOF shares the same mechanism as the pressure-driven flow, the locations of the vortices found in this study are distinct from the typical lip and corner vortices occurring in pressure-driven viscoelastic flows. This is probably because of the different velocity profiles in the pressure-driven flow and the EOF. In pressure-driven flow, the velocity is zero at solid walls and increases to a maximum at the centerline of the geometry. However, the EOF velocity profile is nearly a plug flow as shown in Figure 4b. The velocity increases from zero to a plateau within the EDL thickness, which is only on the order of a few nanometers. For the pressure-driven flow, the highest velocity is at the centerline of the geometry and the velocity near the wall is relatively low, resulting in the stagnant region near the solid boundaries (lips and corners). However, EOF velocity in the vicinity of the charged wall is almost the same as that in the channel centerline. For the extensional flow of viscoelastic fluids, the stretched polymer molecules lead to large elastic stresses, which significantly depend on the geometry and velocity profile. The induced elastic stresses render the primary flow unstable and cause an irregular secondary flow. The flow subsequently acts back on the polymer molecules and stretches them further, causing a strong disturbance of the EOF and yielding a time-dependent EOF.

### 4.2. Elastic Instabilities under Various $E_{app}$ and $c_{p}$

In order to study the effects of $E_{\mathrm{app}}$ and $c_{p}$ on the instabilities of viscoelastic EOF, $c_{p}$ is varied from 100 ppm to 500 ppm and $E_{\mathrm{app}}$ is varied from 100 V/cm to 600 V/cm. Flow patterns under different conditions of $E_{\mathrm{app}}$ and $c_{p}$ are shown in Figure 11, Figure 12, Figure 13 and Figure 14. At certain $E_{\mathrm{app}}$ ($c_{p}$), EOF becomes more unstable with the increase of $c_{p}$ ($E_{\mathrm{app}}$). Figure 11 shows the streamlines for different PAA concentrations under $E_{\mathrm{app}}=600\;V/\mathrm{cm}$, and Figure 12 shows the streamlines within the constriction microchannel with the color bar representing pressure for Newtonian fluid and $\tau_{xx}$ for PAA solutions. As shown in Figure 11 and Figure 12, when $c_{p}$ increases, the polymeric stress $\tau_{xx}$ at the entrance of the constriction microchannel increases rapidly, resulting in the fluctuation of the streamlines at upstream of the microchannel. When $c_{p}$ is relatively low (100 ppm), EOF of viscoelastic fluid is similar to that of Newtonian fluid, and the flow is in a steady state. With an increase in the PAA concentration up to 150 ppm, significant curvature of the centerline streamlines is observed, and the streamlines become asymmetric about the *x*-axis and *y*-axis. As $c_{p}$ continuously increases up to 200 ppm, a pair of upstream vortices in opposite flow directions are induced at upstream of the constriction microchannel, forming a stagnant region as shown in Figure 11d. The width and length of the pair of vortices are about 1.6 time of the constriction microchannel height (i.e., $1.6H_{c}$). Such vortices are found to grow significantly in size with increasing $c_{p}$, which is in qualitative agreement with the experimental observations of Ko [38]. Within the constriction microchannel, as shown in Figure 12d, nearly 1/4 of the microchannel length (i.e., $\frac{1}{4}L_{c}$) from the entrance shows a significant increase of $\tau_{xx}$. Near the downstream lips of the microchannel, a local maximum of the polymeric stress $\tau_{xx}$ is observed. When $c_{p}$ increases to 250 ppm, the fluctuation of the streamlines and the size of the vortices grow dramatically as shown in Figure 11e, in which the width and length of the vortices are about 2.9 times of the microchannel height (i.e., $2.9H_{c}$). The region with significant value of $\tau_{xx}$ is near 1/3 of the microchannel length (i.e., $\frac{1}{3}L_{c}$), as shown in Figure 12e. When $c_{p}$ further increases to 500 ppm, as shown in Figure 11f, the vortices grow into 4.4 times of the constriction microchannel height (i.e., $4.4H_{c}$). More than half of the microchannel length shows a significant increase in $\tau_{xx}$. Furthermore, a small vortex is induced near the downstream lip of the microchannel, which is also reported in experimental studies of Ko [38].

Figure 13 and Figure 14 show the streamlines for $c_{p}=150\;\mathrm{ppm}$ when $E_{\mathrm{app}}$ is varied from 100 V/cm to 600 V/cm with the color bar representing the magnitude of $\tau_{xx}$. At a relatively low electric field such as $E_{\mathrm{app}}=100\;V/\mathrm{cm}$, EOF of PAA solution is similar to the Newtonian fluid, and the flow is in a steady state and symmetric about channel centerline. In addition, the induced polymeric stress $\tau_{xx}$ in the constriction microchannel is relatively small. When $E_{\mathrm{app}}$ increases up to 400 V/cm, centerline streamlines start to show notable fluctuation and become asymmetric about the *x*-axis. At the entrance of the microchannel, a slight increase of $\tau_{xx}$ is observed, however, significant increase of $\tau_{xx}$ is observed near the microchannel walls and the downstream lips, as shown in Figure 14d. When $E_{\mathrm{app}}$ increases to 500 V/cm, a pair of upstream vortices are induced at upstream of the microchannel, and the size of vortices is about the height of constriction microchannel. Additionally, as shown in Figure 14e, significant $\tau_{xx}$ is induced near the entrance of the constriction microchannel. However, when $E_{\mathrm{app}}$ further increases to 600 V/cm, the size of the upstream vortices do not show notable increase in comparison with that of $E_{\mathrm{app}}=500\;V/\mathrm{cm}$. The results clearly show that increase of $c_{p}$ and/or $E_{\mathrm{app}}$ can magnify the elastic instabilities of the viscoelastic EOF. However, the increase of $c_{p}$ has a more significant enhancing effect on the elastic instabilities of the viscoelastic EOF than the increase of $E_{\mathrm{app}}$.

Figure 15 depicts a flow map for the onset of vortices in unstable EOF as functions of $c_{p}$ and $E_{\mathrm{app}}$. At a fixed $c_{p}$ ($E_{\mathrm{app}}$), vortices and unstable EOF occur when $E_{\mathrm{app}}$ ($c_{p}$) exceeds a certain threshold value. For example, for $c_{p}=200\;\mathrm{ppm}$, the flow becomes unstable with the occurrence of vortices when $E_{\mathrm{app}}$ is above 300 V/cm. At relatively low PAA concentration (i.e., $c_{p}=100\;\mathrm{ppm}$), it requires a very high electric field (up to 850 V/cm) to yield unstable EOF with upstream vortices. In contrast, at relatively high $c_{p}$ (i.e., $c_{p}=500\;\mathrm{ppm}$), the onset of vortices occurs at $E_{\mathrm{app}}$ between 50 V/cm and 100 V/cm. An asymptotic curve fitting is implemented to illustrate the transition condition from no upstream vortices to the formation of upstream vortices, which is given as $E_{\mathrm{app}}=47.49+2892.25\cdot 0.987^{c_{p}}$, where $c_{p}$ represents the polymer concentration in ppm, and $E_{\mathrm{app}}$ is the apparent electric field in V/cm. Above the curve in Figure 15, the EOF becomes time-dependent with upstream vortices, and no vortex forms under the conditions below the curve. Note that the flow map is only valid for the geometry considered in this study with the zeta potential of −110 mV. The instabilities of the viscoelastic EOF are dependent on the value of zeta potential. A comparison of the flow patterns of lower zeta potential (−70 mV) and higher zeta potential (−150 mV) for 150 ppm PAA solution under $E_{\mathrm{app}}=$ 600 V/cm in the Appendix B clearly shows that higher zeta potential triggers stronger instabilities under the same condition. In addition, the dimensionless numbers (Reynolds number and Weissenberg number) of the studied conditions corresponding to Figure 15 are given in the Appendix B.

Since the flow velocity is time-dependent, we first calculate the cross-sectional average velocity over a period of $\Delta t=t_{2}-t_{1}$, and then take the time-average to obtain the averaged velocity as,
(11)$$\bar{U}=\frac{\int_{t_{1}}^{t_{2}}\int_{-H_{c}/2}^{H_{c}/2}U\left(0,y\right)dydt}{\Delta t\cdot H_{c}},$$

We choose Δt=1 s in the current study. Figure 16a shows the average velocity at the center of the constriction as a function of $c_{p}$ at different values of $E_{\mathrm{app}}$. In comparison, it also shows the result of Newtonian fluid whose viscosity is the same as the total viscosity of PAA solution under $E_{\mathrm{app}}=600\;V/\mathrm{cm}$. Under the same $E_{\mathrm{app}}=600\;V/\mathrm{cm}$, the average velocity in the constriction microchannel of the Newtonian flow is about 6–12% higher than that of the PAA solution. The decrease of the average velocity in PAA solution is attributed to the induced polymeric stress at the entrance of the constriction microchannel. For Newtonian fluid, the average velocity decreases as $c_{p}$ increases, which is due to the increase of viscosity to make its viscosity be the same as that of PAA solution with the concentration of $c_{p}$. For PAA solutions, under the same $E_{\mathrm{app}}$ the average velocity exponentially decreases as the polymer concentration increases. One reason is attributed to the increase of total viscosity with the increase in $c_{p}$. In addition, the induced polymer stress within the constriction increases with the increase of polymer concentration, as shown in Figure 12, and the induced polymer stress slows down the flow.

Under the considered condition of $c_{0}=0.01\;\mathrm{mM}$, the EDL thickness is only 95 nm, which is much smaller than the height of the constriction. In Newtonian fluid, the EOF velocity can be approximated by the well-known Helmholtz–Smoluchowski velocity formula as described in Equation (10). We wonder if the average velocity of PAA solutions can be still approximated with the Helmholtz–Smoluchowski velocity formula. Since the actual local electric field within the constriction is much higher than the apparent electric field $E_{\mathrm{app}}$, time-averaged electric field in the x-direction at the center of the constriction is used in the calculation of the Helmholtz–Smoluchowski velocity. Figure 16b shows the average velocity as a function of the apparent electric field under various PAA concentrations. The lines in Figure 16b represent the corresponding EOF velocity predicted by the Helmholtz–Smoluchowski velocity formula. At a fixed $c_{p}$, as expected, the EOF velocity increases with an increase in the applied electric field. In general, the Helmholtz–Smoluchowski formula over predicts the velocity, and at a fixed $c_{p}$ the relative error increases with the increasing $E_{\mathrm{app}}$. For example, for $c_{p}=100\;\mathrm{ppm}$, the relative errors under $E_{\mathrm{app}}=$100 V/cm and 600 V/cm are, respectively, 1.2% and 8.6%. At a fixed $E_{\mathrm{app}}$, the absolute error, which is the difference between the Helmholtz–Smoluchowski approximated velocity and the average velocity obtained from the full numerical simulation, increases with the increasing PAA concentration. However, the relative error shows no notable change with the increasing $c_{p}$. For example, at $E_{\mathrm{app}}=600\;V/\mathrm{cm}$, the relative errors for $c_{p}=100\;\mathrm{ppm},\;300\;\mathrm{ppm},\;\mathrm{and}\;500\;\mathrm{ppm}$ are, respectively, 8.7%, 9.6%, and 9.0%. To evaluate the applicability of the Helmholtz–Smoluchowski formula to approximate the velocity of viscoelastic fluids for $c_{p}$ ranging from 100 ppm to 500 ppm, the minimum, average, and maximum relative errors at different $E_{\mathrm{app}}$ are calculated as shown in Table 1. When $E_{\mathrm{app}}\leq 300\;V/\mathrm{cm}$, the relative error is less than 5%. However, when $E_{\mathrm{app}}\geq 400\;V/\mathrm{cm}$, the relative error is larger than 5%, and the Helmholtz–Smoluchowski formula failed to predict the velocity of the viscoelastic fluids accurately. In this study, the largest relative error is 9.6% when $c_{p}=300\;\mathrm{ppm}$ and $E_{\mathrm{app}}=600\;V/\mathrm{cm}$.

## 5. Conclusions

Electroosmotic flow (EOF) of viscoelastic fluid through a 10:1:10 constriction microchannel is numerically investigated as functions of the applied electric field and the polymer concentration. In the current study, we neglect the effect of the polymer concentration on the zeta potential of the channel walls. Comparing to the EOF of Newtonian fluid, the following distinct results for viscoelastic EOF through a 10:1:10 constriction microchannel are obtained:(1)When polyacrylamide (PAA) concentration (applied electric field) exceeds a critical value, the EOF of viscoelastic fluid becomes time-dependent with upstream vortices occurring in the inlet reservoir near the entrance of the constriction microchannel. In contrast, EOF of Newtonian fluid is always in a steady state without vortices.(2)For the viscoelastic EOF, significant polymer stress is induced near the entrance within the constriction and near the downstream lips of the constriction, causing the elastic instabilities of the viscoelastic EOF. The induced polymer stress is dramatically magnified with the increase of polymer concentration and applied electric field. However, the increase of polymer concentration shows a more significant enhancing effect on the polymer stress than the increase of applied electric field.(3)The EOF velocity of viscoelastic fluid within the constriction becomes temporally and spatially dependent. Near the exit of the constriction, due to the extrudate swell effect of the polymers, the velocity at the centerline first decreases at the exit followed by an increase in the outlet reservoir.(4)The velocity at the exit of the constriction is higher than that at the entrance of the constriction because of the formation of upstream vortices, which is in qualitative agreement with experimental observation obtained from the literature.(5)Under the same total viscosity and applied electric field, the velocity of Newtonian fluid is higher than that of viscoelastic fluid, which is attributed to the induced polymeric stress within the constriction. When the applied electric field is less than 300 V/cm, the Helmholtz–Smoluchowski velocity formula can predict the cross-sectional average velocity of viscoelastic fluid with PAA concentration up to 500 ppm, and the relative error is less than 5%. At a fixed PAA concentration, in general the relative error of the Helmholtz–Smoluchowski approximation increases with an increase in the applied electric field.

## Figures and Tables

**Figure 1 micromachines-12-00417-f001:**
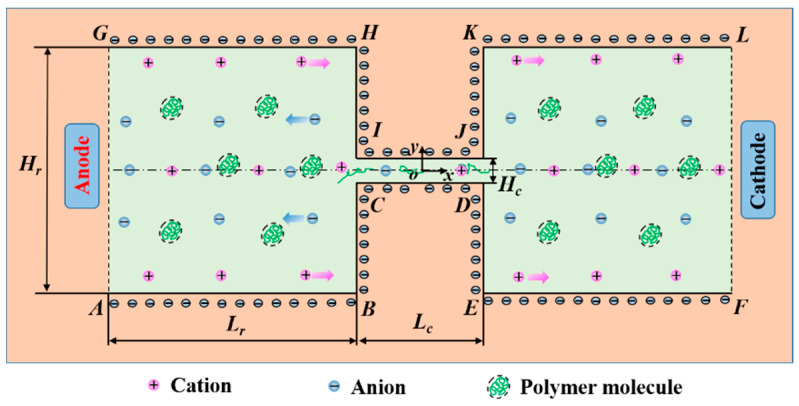
Schematic diagram of a constriction microchannel connecting two reservoirs at both ends. The solid walls of reservoirs and the constriction channel are negatively charged, and an electric field is imposed by applying a potential difference between anode and cathode positioned in two fluid reservoirs.

**Figure 2 micromachines-12-00417-f002:**
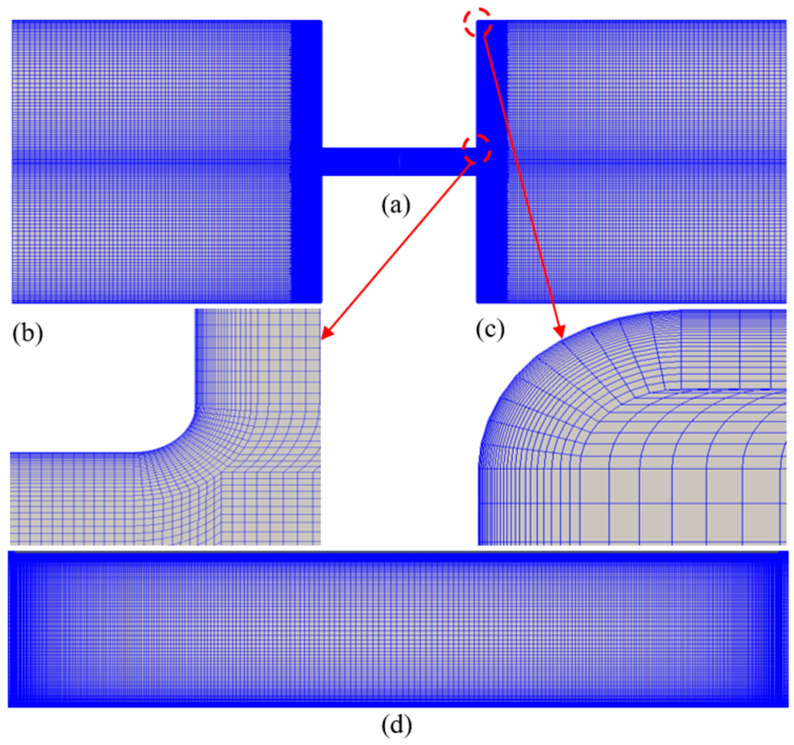
Computational mesh used in the numerical simulations. Mesh of the whole geometry (**a**) and detailed view of the mesh at channel corner (**b**), at reservoir corner (**c**), and in the constriction microchannel (**d**).

**Figure 3 micromachines-12-00417-f003:**
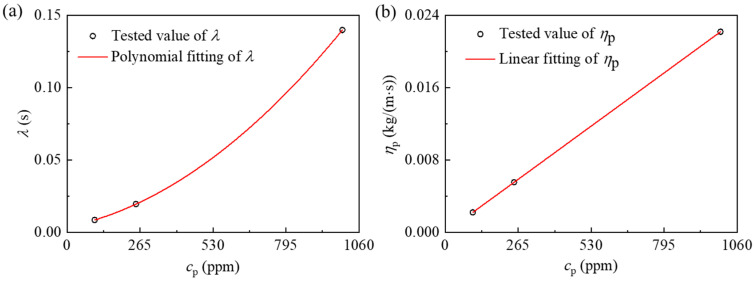
(**a**) Polymer dynamic viscosity $\eta_{p}$ and (**b**) relaxation time λ as a function of the polyacrylamide (PAA) concentration, $c_{p}$.

**Figure 4 micromachines-12-00417-f004:**
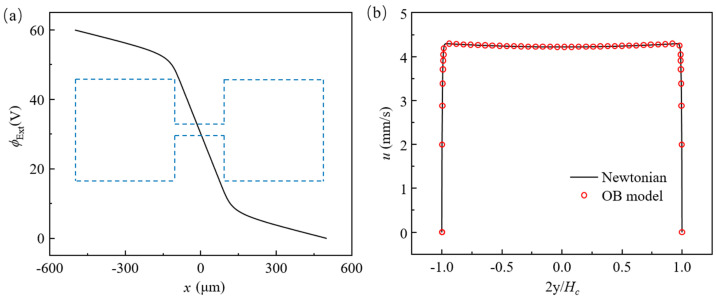
(**a**) Electric potential distribution (blue dash line shows the relative position of the geometry) along the *x*-axis; (**b**) the *x*-component velocity at the center of the constriction microchannel, u(0,y), for Newtonian model (solid line) and OB model (symbol).

**Figure 5 micromachines-12-00417-f005:**
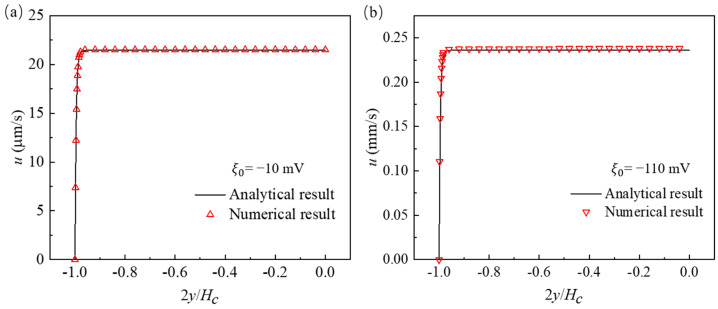
The *x*-component velocity profile of viscoelastic electroosmotic flow (EOF) between two parallel plates: (**a**) Zeta potential is −10 mV; (**b**) zeta potential is −110 mV. Analytical result of Afonso et al. [28] (solid line) and current numerical result (symbol). The analytical solution is described in Appendix B.

**Figure 6 micromachines-12-00417-f006:**
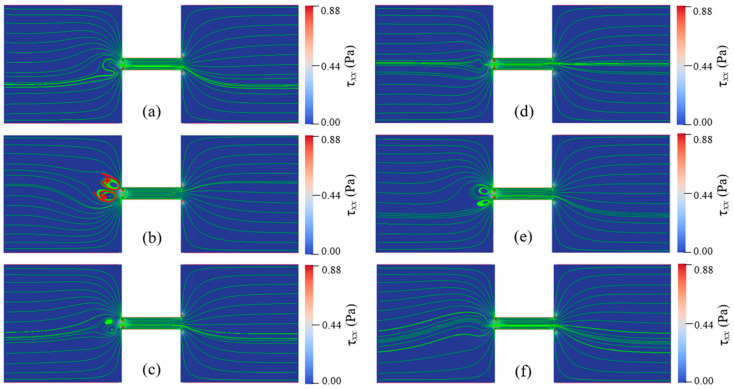
Instability of EOF with $c_{p}=500\;\mathrm{ppm}$ and $E_{\mathrm{app}}=100\;V/\mathrm{cm}$. Streamlines at different times: (**a**) 1.71 s, (**b**) 1.75 s, (**c**) 1.79 s, (**d**) 1.83 s, (**e**) 1.87 s, and (**f**) 1.91 s. The color bar represents the elastic normal stress $\tau_{xx}$. (Appendix A).

**Figure 7 micromachines-12-00417-f007:**
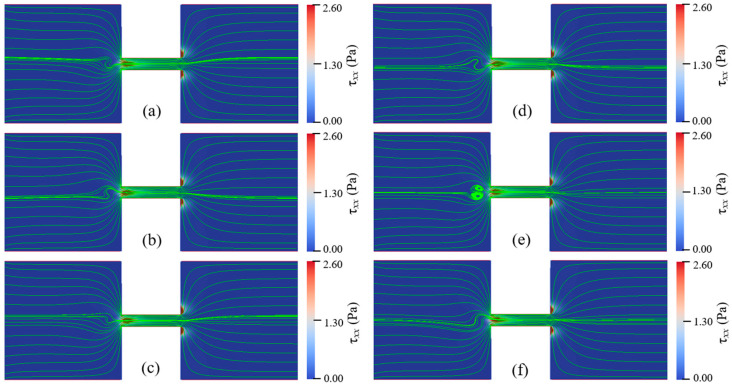
Instability of EOF with $c_{p}=150\;\mathrm{ppm}$ and $E_{\mathrm{app}}=600\;V/\mathrm{cm}$. Streamlines at different times: (**a**) 1.70 s, (**b**) 1.72 s, (**c**) 1.74 s, (**d**) 1.76 s, (**e**) 1.78 s, and (**f**) 1.80 s. The color bar represents the elastic normal stress $\tau_{xx}$. (Appendix A).

**Figure 8 micromachines-12-00417-f008:**
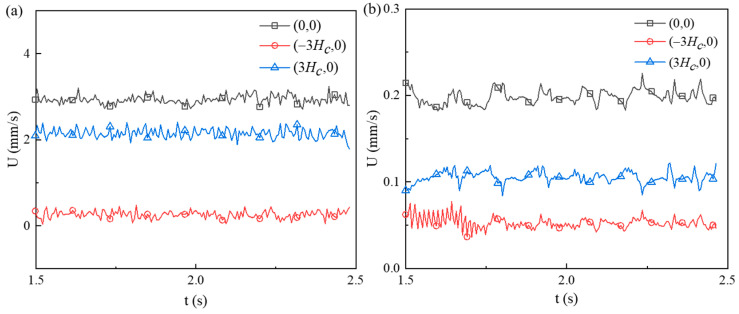
Velocity magnitudes at three different locations ($\left(-3H_{c},0\right)$, (0,0), $\left(3H_{c},0\right)).$ (**a**) $c_{p}=150\;\mathrm{ppm}$ and $E_{\mathrm{app}}=600\;V/\mathrm{cm}$, (**b**) $c_{p}=500\;\mathrm{ppm}$ and $E_{\mathrm{app}}=100\;V/\mathrm{cm}$.

**Figure 9 micromachines-12-00417-f009:**
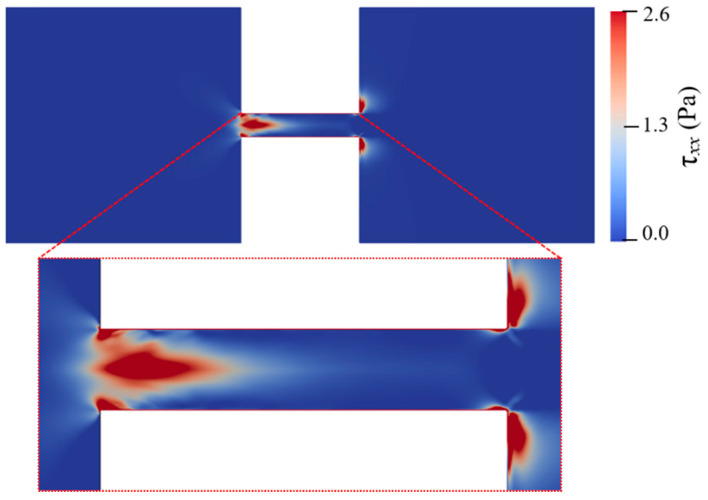
Spatial distribution of the elastic normal stress $\tau_{xx}$ for $c_{p}=150\;\mathrm{ppm}$ and $E_{\mathrm{app}}=600\;V/\mathrm{cm}$ at t=1.78 s.

**Figure 10 micromachines-12-00417-f010:**
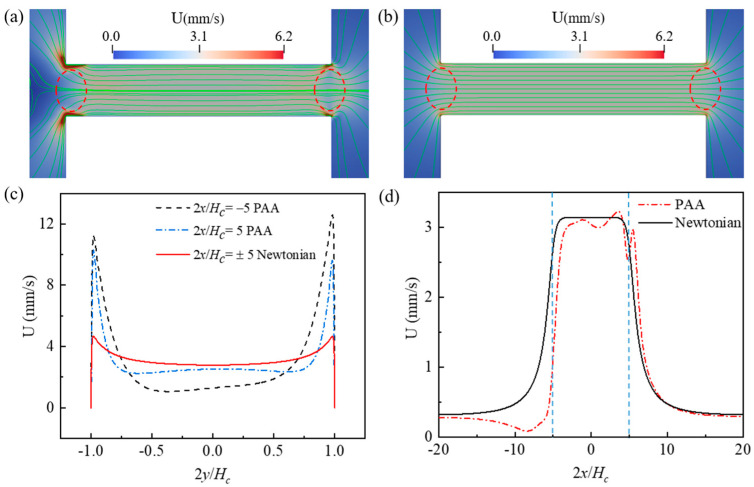
Streamlines and velocity magnitude for $c_{p}=150\;\mathrm{ppm}$ and $E_{\mathrm{app}}=600\;V/\mathrm{cm}$ and Newtonian fluid at t=1.78 s: (**a**) 150 ppm PAA solution, (**b**) Newtonian fluid with same total viscosity as 150 ppm PAA solution, (**c**) velocity magnitude profiles at $2x/H_{c}=\pm 5$, (**d**) velocity magnitudes profiles at y=0 (The blue dash lines show the position of the contraction microchannel). The color bar represents the velocity magnitude U.

**Figure 11 micromachines-12-00417-f011:**
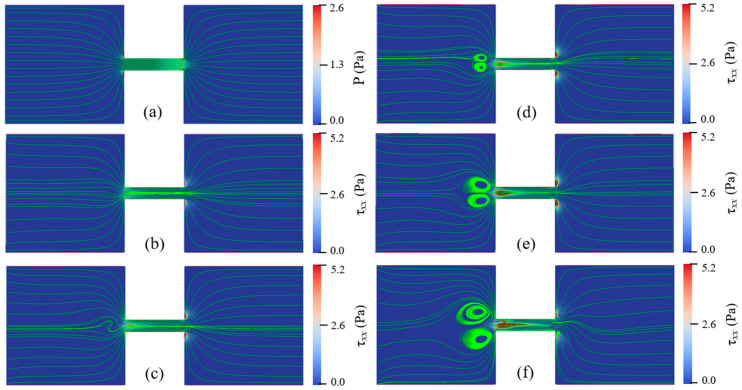
Streamlines of Newtonian fluid and PAA solutions with different concentrations under $E_{\mathrm{app}}=600\;V/\mathrm{cm}$ at 1.70 s: (**a**) Newtonian fluid, (**b**) $c_{p}=100\;\mathrm{ppm}$, (**c**) $c_{p}=150\;\mathrm{ppm}$, (**d**) $c_{p}=200\;\mathrm{ppm}$, (**e**) $c_{p}=250\;\mathrm{ppm}$, and (**f**) $c_{p}=500\;\mathrm{ppm}$. The color bar represents the elastic normal stress $\tau_{xx}$.

**Figure 12 micromachines-12-00417-f012:**
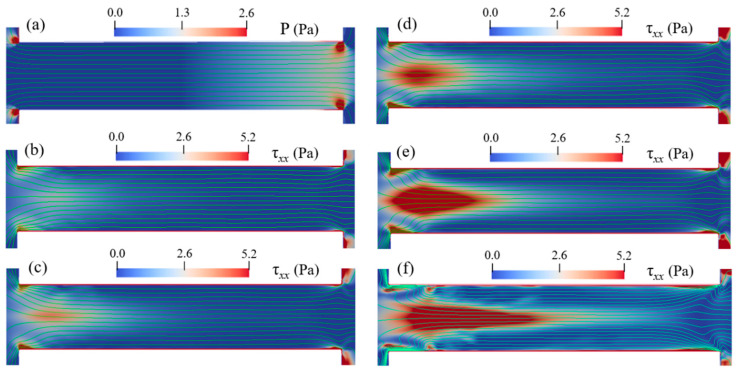
Streamlines in microchannel of Newtonian fluid and PAA solutions with different concentrations under $E_{\mathrm{app}}=600\;V/\mathrm{cm}$ at 1.7 s: (**a**) Newtonian fluid, (**b**) $c_{p}=100\;\mathrm{ppm}$, (**c**) $c_{p}=150\;\mathrm{ppm}$, (**d**) $c_{p}=200\;\mathrm{ppm}$, (**e**) $c_{p}=250\;\mathrm{ppm}$, and (**f**) $c_{p}=500\;\mathrm{ppm}$. The color bar represents the elastic normal stress $\tau_{xx}$.

**Figure 13 micromachines-12-00417-f013:**
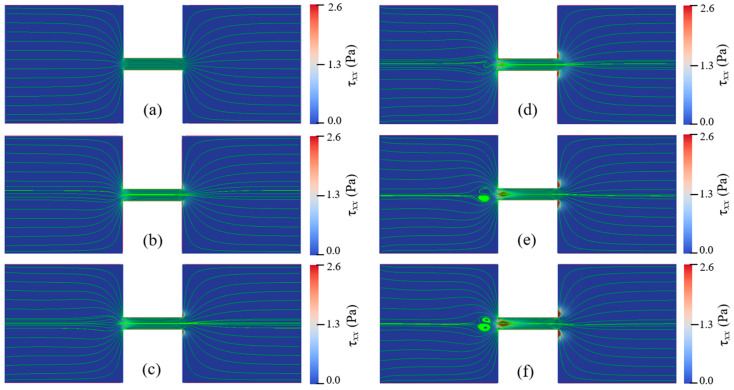
Streamlines for 150 ppm PAA solution under different $E_{\mathrm{app}}$ at 1.78 s: (**a**) 100 V/cm, (**b**) 200 V/cm. (**c**) 300 V/cm, (**d**) 400 V/cm, (**e**) 500 V/cm, (**f**) 600 V/cm. The color bar represents the elastic normal stress $\tau_{xx}$.

**Figure 14 micromachines-12-00417-f014:**
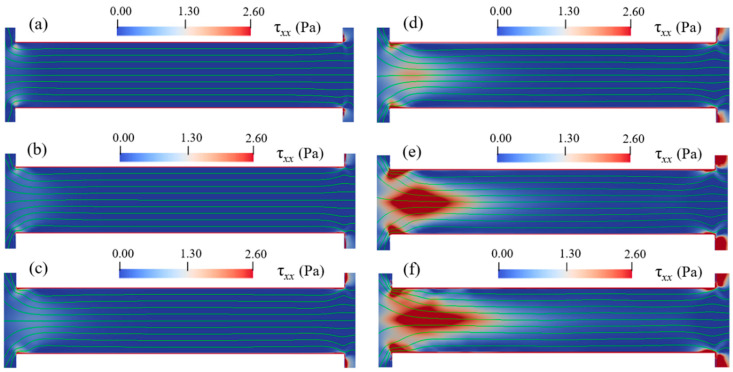
Streamlines in microchannel of 150 ppm PAA solution under different $E_{\mathrm{app}}$ at 1.78 s: (**a**) 100 V/cm, (**b**) 200 V/cm. (**c**) 300 V/cm, (**d**) 400 V/cm, (**e**) 500 V/cm, and (**f**) 600 V/cm. The color bar represents the elastic normal stress $\tau_{xx}$.

**Figure 15 micromachines-12-00417-f015:**
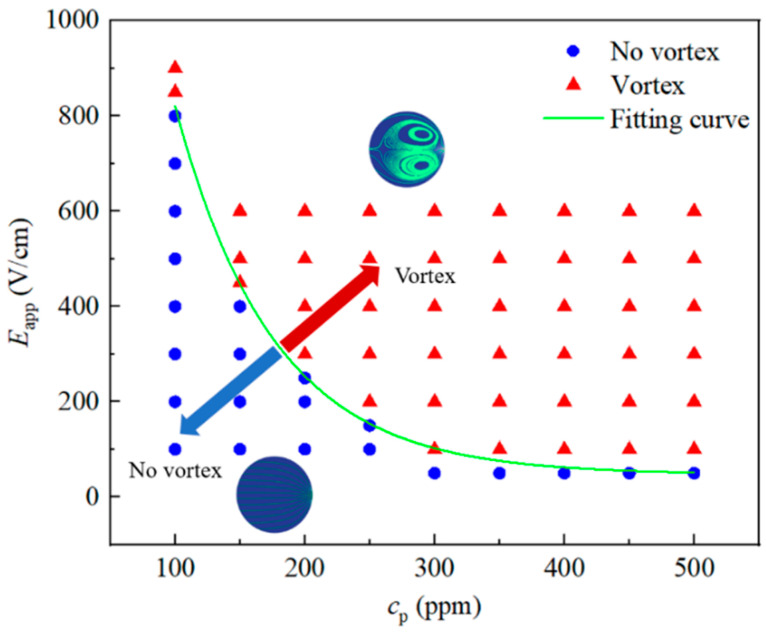
Flow map in $c_{p}$-$E_{\mathrm{app}}$ space for EOF of PAA solutions flowing through a 10:1:10 constriction/expansion microchannel. Up-right of the fitting curve are the conditions that trigger the vortex in the EOF.

**Figure 16 micromachines-12-00417-f016:**
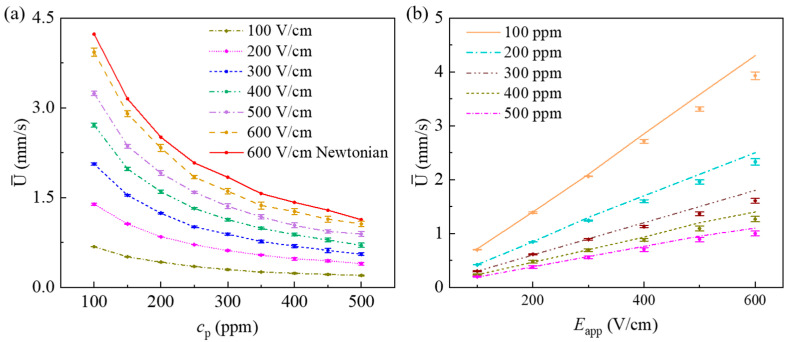
Time averaged cross-sectional average velocity at the center of the constriction microchannel (*x* = 0): (**a**) Average velocity, (**b**) comparison of average velocity (with deviation) and Helmholtz–Smoluchowski velocity (lines).

**Table 1 micromachines-12-00417-t001:** Relative error between the Helmholtz–Smoluchowski velocity and the average velocity from the full mathematical model.

$E_{\mathrm{app}}(V/\mathrm{cm})$	100	200	300	400	500	600
Minimum relative error	1.0%	1.3%	1.8%	5.5%	7.0%	8.5%
Average relative error	1.3%	1.6%	2.3%	5.9%	7.7%	8.9%
Maximum relative error	1.5%	1.9%	2.8%	6.5%	8.2%	9.6%

## Data Availability

Not applicable.

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
