# Peer review of "Electroosmotic Flow of Viscoelastic Fluid through a Constriction Microchannel"

_micromachines, 2021, doi:10.3390/mi12040417_

Round 1
Reviewer 1 Report
Review of Manuscript ID: # micromachines-1121146 titled: "Electroosmotic Flow of Viscoelastic Fluid through a Con-2 striction Microchannel" submitted to micromachines.
The manuscript presents a computational study of electroosmotic flow for a non-Newtonian fluid in a constriction microchannel. The manuscript shows the effect of vortex generation as a function of the apparent electric field and PAA polymer solution concentration for a single geometry and zeta potential. However, the manuscripts lack a rigorous numerical validation, the mesh independence study is not provided, the findings are narrow to the specific proposed geometry/conditions, and the results and discussion section is poorly structured. I do not recommend the manuscript for publication. I hope the following comments may be useful for future submissions.
Line numbers correspond to those generated by micromachines.
Major comments.
1) For Newtonian fluids, there is extensive literature on electrokinetic instabilities (EKI). For instance, in the work by Posner and Santiago [1], it was demonstrated that EKI is controlled by a modified local electric Rayleigh which is a function of the depth (3rd dimension) (see eq 3.19 page 10). In lines 128-130, the manuscript assumes a 2D simulation based on W (width) >> H (height), L (large). However, in microfluidics applications (the motivation provided in the introduction of the manuscript in lines 32-34) the width is usually comparable to height, which invalidated the 2D assumption. Therefore, I would think that a 3D simulation could be more appropriate to fully characterize the problem. If, however, the manuscript wants to focus on a 2D case, a few 3D simulations should be provided to verify the 2D assumption. Please clarify and provide 3D simulations.
[1] Posner and Santiago “Convective instability of electrokinetic flows in a cross-shaped microchannel”, J. Fluid Mech., 2006
2) In line 33, the manuscript motivated the work with proteins and DNA applications. In such applications, it is common to use buffers, weak electrolytes, to control the pH. Why only strong electrolytes are being considered in the manuscript? The scope of the work should be reflected in the motivation in the introduction. Otherwise, expanding the simulation to include weak electrolytes must be included.
3) The manuscript uses the Oldroyd-B (OB) constitutive model. In lines 43-35, the manuscript provides a list of another constitutive model. However, in lines 145-146, the manuscript mentions the use of the OB model without further discussion. Please provide a discussion with the advantages and disadvantages of the models listed in lines 43-45. The information could be summarized in a table.
4) In lines 275-277, if I understand correctly, the OB model was used with parameters near a Newtonian case to compare both cases. However, the manuscript concludes in line 277 that this is sufficient to validate the numerical code. Unfortunately, this only validates that the OB model correctly predicts a Newtonian case. This not necessarily means that the OB model is validated for any other simulation parameters. Please provide a more robust validation of the OB model with experimental data (if available in the literature) or numerical results in regimes closer to the simulations presented in the manuscript.
5) In lines 563-565, the manuscript comments that the flow map (Figure 14) is only valid for the current geometry and zeta potential of -110 mV. This narrows the applicability of the paper; therefore, more zeta potentials are necessary to find a more generalizable conclusion of the paper. In other words, since Figure 14 is the key figure of the manuscript, this Figure should be generated for more cases to ideally find a non-dimensional parameter to explain the data. Please, at least, provide more simulation at different zeta potentials.
6) In lines 121-123, the manuscript neglects the effect of the polymer concentration on the wall zeta potential. Please provide a range of concentrations that makes this assumption valid and support the claim with an order of magnitude analysis from a governing equation.
7) In the introduction, the manuscript mentions the limited experimental and numerical investigation regarding EOF instabilities of viscoelastic fluid. Since the available literature is limited, please provide a comprehensive discussion of the findings of these works. This information could be summarized in a table and should include the geometries type, electrolyte, range of employed parameters, comments about instabilities, etc. To avoid confusion in the manuscript, I suggest including this information in a new subsection in the results and discussion. This information will help to understand the contribution of the manuscript.
8) Regarding λ, the relaxation time of the polymer. Please provide a short physical insight description of this term, and the method used to have a reliable measurement/estimate of it.
9) In lines 347-358, the manuscript describes the work from Sadek et al. However, this work is in a different geometry and the paragraph, itself, combines several ideas. I recommend creating a subsection to exclusively explain the finding in the literature regarding instabilities. See comment 7 above.
10) In lines 224-225 the manuscript uses a fillet of 2 μm to avoid sharp turns. Did the manuscript considerer other fillet diameters? What is the effect of the fillet on the instabilities? Please provide a few simulations in a near instability and instability regime at different fillet sizes.
11) The structure of the results section is confusing. For example, in lines 283-289, the introductory paragraph for the discussion of the results starts with results, followed by the approach, and then provides some findings. After this, a description of the section overview is provided. I found this way of structuring the paragraph very confusing.
12) In line 232, the manuscript claims that a mesh independence study was performed. Where is this study? Please provide a figure and discussion about this in the SI.
13) In lines 327-336, the manuscript provides a description of the structures generated. Videos with explanations could be provided in the SI to visualize the development of the vortices.
Comments
14) Overall, the caption of the figures must be improved. More details should be provided.
15) Line 150: the manuscript mentions a “relatively high Weissenberg number”, specify a range.
16) Line 163: the manuscript says that “EDL thickness is on the order of nanometers”. Please show some calculations to support this claim, it could be the Debye length.
17) In lines 171-178, the manuscript describes the boundary conditions (BCs). For smoother readability, please add the BCs in figure 1 or a new figure in the SI.
18) Line 237: I noted that eta and lambda were obtained from reference 49. Are these parameters experimentally calculated? Please include a comment.
19) Lines 253-255: the punctuation of the sentence is confusing.
20) line 255: why a lambda of 0.1 ms was chosen? Please provide a comment on the reasoning.
21) In lines 296-310, this paragraph looks to be intended as a new introductory paragraph to a subsection. I find this confusing and repetitive.
22) Lines 323-324: The sentence starts with an independent clause and there is no dependent clause to properly finalize the sentence.
23) Line 327: missing comma after "Then".
34) Line 499: “?p is relatively low”, please be specific about the range.
Author Response
See the response to reviewers.

Reviewer 2 Report
The manuscript is generally well written and the experimental work rationally described. I have just few suggestions which may improve the quality:
- showing shear stress as a function of shear rate would be helpful to visualize fluid rheologic behaviour;
- comparing computational results with experimental characterisation of fluid viscoelasticity would be interesting. Lumped parameter models are also often used for describing viscoelastic fluids (see for example https://link.springer.com/article/10.1140/epje/i2015-15033-4, https://www.sciencedirect.com/science/article/pii/S2589152919303485). Maybe these points can be discussed and proposed as a further developments.
- In Figure 9 caption: S Streamlines
Author Response
See the response to reviewers.

Round 2
Reviewer 1 Report
The manuscript presents a computational study of electroosmotic flow for a non-Newtonian fluid in a constriction microchannel. The manuscript shows the effect of vortex generation as a function of the apparent electric field and PAA polymer solution concentration for a single geometry and zeta potential. The responses of the first revision show that several of the relevant comments were not addressed including, a rigorous numerical validation, the mesh independence study is not provided, the findings are narrow to the specific proposed geometry/conditions, and several references are missing to support the claims. In its current form, I do not recommend the manuscript for publication. I hope the following comments may be useful for future submissions.
R2-1) The response to my first comment (1) of the first revision claims that there is relevant literature in microfluidics where the width is much larger than the height. Please provide several references in the introduction and 2D assumption to support this claim.
R2-2) To clarify my first comment (1) of the first revision, in electrokinetic instabilities (EKI) for Newtonian fluids there are relevant non-dimensional numbers to characterize the problem. It would be beneficial to this work to see such types of parameters for a few relevant cases. A well-characterized non-dimensional number provides much more information.
R2-3) The response to my second comment (2) of the first revision claims “typically the concentration of the weak electrolyte is relatively low comparing with that of the background salt solution”. Please, add this discussion in the manuscript with the corresponding references.
R2-4) The response to my fourth comment (4) of the first revision claims that the validation of the code was benchmarked with an analytical solution for a low zeta potential case. However, the employed zeta potential in the manuscript is 11 times larger than the zeta potential used for validation. Please use a zeta potential comparable to -110 mV for the validation case.
R2-5) The response to my fourth comment (4) of the first revision claims that there is no experimental data available to compare with their work. After a quick online search, I was able to find some work regarding experimental and some interesting simple numerical validations:
-Cho et al. [1] compared the velocity distributions for a Newtonian and non-Newtonian case (power-law with n = 0.5).
-Zhao et al. [2] provides an analytical solution for multiple power-law fluids.
-Bag and Bhattacharyya [3] showed a comparison with Zhao’s power-law flow solutions and experimental data from Huang et al. [4].
I noticed that the new version of the manuscript includes a comparison with Alfonso et al. Please provide the equation and cite Zhao [2] (if it is the same equation). Also, please perform a comparison as developed by Cho et al. [1] to see if the model captures the differences between Newtonian and non-Newtonian fluids. And finally, please compare the results with the experimental data from Huang et al. [4]
R2-6) The response to my fifth comment (5) of the first revision claims that “this manuscript is already very long with the consideration of two parameters: electric field strength and PAA concentration”. If space is an issue, the data could be either added as a figure with multiple a plots a, b, c, etc, or the extra data can be added in the SI.
R2-7) The response to my sixth comment (6) of the first revision claims that “there is no data available from the literature to show the relationship between the polymer concentration and zeta potential”. After a quick online search, the paper by Huang et al [4] shows in figure 7 the zeta potential for several concentrations of PEO and electric fields. Please, find more related work and provide the proper references.
R2-8) The response to my tenth comment (10) of the first revision claims that “Simulation of fillets with different sizes will be conducted in our future studies”. The same response also claims: “the small fillets will not affect the instability observed from our results”. Therefore, if it is expected a negligible effect of fillets, there is no need to explore this in a future study. However, it is important to sustain this claim with a couple of simulations. Please provide these simulations in the SI.
R2-9) The response to my twelfth comment (12) of the first revision claims that “We performed mesh independence study and added the related results in support information”. However, I only found 2 videos and no SI text file.
R2-10) There are two videos provided in the SI. Please, add text in the video to indicate the flow properties, add information with the simulation time (and delta t), please increase the frames per second and length of the video, and also decrease delta t (it is hard to see the vortex formation).
References
[1] Cho et al., Electrokinetically-driven non-Newtonian fluid flow in rough microchannel with complex-wavy surface, J. of Non-Newtonian Fluid Mechanics, 2012
[2] Zhao et al., Analysis of electroosmotic flow of power-law fluids in a slit microchannel, J. of Colloid and Interface Sc., 2008
[3] Bag and Bhattacharyya, Electroosmotic flow of a non-Newtonian fluid in a microchannel with heterogeneous surface potential, J. of Non-Newtonian Fluid Mechanics, 2018
[4] Huang et al., Experimental and theoretical investigations of non-Newtonian electro-osmotic driven flow in rectangular microchannels, Soft Matter, 2016
Round 3
Reviewer 1 Report
The authors addressed properly the comments and the paper can be considered for publication.